# Optical Polarimetric Detection for Dental Hard Tissue Diseases Characterization

**DOI:** 10.3390/s19224971

**Published:** 2019-11-14

**Authors:** Tien-Yu Hsiao, Shyh-Yuan Lee, Chia-Wei Sun

**Affiliations:** 1Biomedical Optical Imaging Lab, Department of Photonics, College of Electrical and Computer Engineering, National Chiao Tung University, No.1001, University Road, East District, Hsinchu City 30010, Taiwan; u09900139@gmail.com; 2Department of Stomatology, Taipei Veterans General Hospital, No. 201, Section 2, Shipai Road, Beitou District, Taipei City 11217, Taiwan; sylee@ym.edu.tw; 3School of Dentistry, National Yang-Ming University, No. 155, Section 2, Linong Street, Beitou District, Taipei City 11221, Taiwan

**Keywords:** optical polarimetric detection, dental diseases

## Abstract

Dental enamel constitutes the outer layer of a crown of teeth and grows nearly parallel. This unique nanostructure makes enamel possess birefringence properties. Currently, there is still no appropriate clinical solution to examine dental hard tissue diseases. Therefore, we developed an optical polarization imaging system for diagnosing dental calculus, caries, and cracked tooth syndrome. By obtaining Stokes signals reflected from samples, Mueller images were constructed and analyzed using Lu-Chipman decomposition. The results showed that diattenuation and linear retardance images can distinguish abnormal tissues. Our result also aligns with previous studies assessed by other methods. Polarimetric imaging is promising for real-time diagnosing.

## 1. Introduction

Dental enamel constitutes the outer layer of the crown of teeth and is the most highly mineralized and durable material in the human body, which provides the inner layer of teeth protection from external mechanical wearing, chemical etching, and the temperature stimulus [1]. The anisotropic nanostructure of enamel determined by the secretory process of ameloblasts the so-called Tomes’ process—is the key to the hardness of teeth [2]. During the Tomes’ process, a group of ameloblasts secrete hydroxyapatite crystals, which grow with a nearly parallel crystalline orientation and are cemented into a bundle to enhance the strength and bendability [3]. The crystal bundles are known as the enamel prisms, which have keyhole-shaped cross-sections, and radiate out from the dentinoenamel zone to the surface, composing the entire crown of the tooth [4]. Unlike other tissues, enamel does not regenerate once the formation of the tooth is finished. Therefore, if the tooth is infected with dental hard tissue diseases, the enamel is permanently damaged and changes occur in its structural property. Thus, examining the changes of the enamel nanostructure is considered potentially useful not only for research on the progression of teeth hardening but also for clinical diagnosis of dental hard tissue diseases.

Dental diseases are inevitable and common problems that not only deteriorate quality of life but also threaten human health. Dental diseases, such as periodontitis, cracked tooth syndrome, and dental caries, cause tooth loss, problems with eating, and even other induced comorbidities [5,6]. Dental caries and periodontitis are two of the most severe global dental problems [7,8]. There are ways to diagnosis dental caries, including a visual-tactile method [9,10], radiography [11], caries detection dyes [12], etc [13,14,15]. However, none of these methods can assess changes of the tooth nanostructure quantitatively and clinically at the same time. Dental calculus is a type of mineralized plaque, which is an etiological factor of periodontitis. Diagnosis of calculus is traditionally based on clinical examination by periodontal probing and radiography. Nonetheless, diagnosis using periodontal probing depends on, to some extent, the amount of experience the doctor has and this procedure has poor reproducibility. Radiographic imaging also has low specificity in the case of the calculus being located on the lingual or buccal sides [11]. Although several methods, such as using smart ultrasonic devices [16], light-emitting diode-based optical probing [17], and laser-induced fluorescence [18,19], have been developed for detecting a dental calculus [20], the sensitivity and reliability are still insufficient for clinical practice. Cracked tooth syndrome is defined as an incomplete fracture of the tooth. Due to the difficulty of visualization and the variety of symptoms depending on its morphology, cracked tooth syndrome is one of the trickiest dental diseases even for the most experienced dentists to diagnosis [21]. Notwithstanding the clinical examinations of cracked tooth syndrome including periodontal probing, radiography, bite test, transillumination, etc [22], diagnostic tools with better reliability are still largely needed.

Thanks to the unique nanostructure of the enamel, the tooth surface exhibits an optical birefringence property [23], which means that the intensity response of the reflected light changes with a different polarization orientation of the incident light. It has been reported that the observed enamel birefringence is the sum of the intrinsic (related to hydroxyapatite crystals and with negative sign) and the form (related to non-mineral volumes and with positive sign) birefringence [24]. When a polarized light is emitted on a tooth, the polarization state of the backscatter light from the infected part is expected to differ from the uninfected part because of the nanostructural changes arising from dental hard tissue diseases. By observing the polarization states images, a lesion can be distinguished quantitatively. There have been dozens of studies utilizing polarized light microscopy (PLM) and polarization sensitive optical coherence tomography (PS-OCT) to investigate the birefringence characteristics of the tooth [24,25,26] and to monitor the progress of dental caries [27,28,29] as well as remineralization [30,31] and demineralization processes [32]. Previous research studies corroborated the applicability of polarimetric detection for carious teeth examination. Nonetheless, these studies only focused on dental caries whereas none of the other dental hard tissue diseases were discussed. In addition, the true meaning of indices used were still ambiguous in account of diattenuation and depolarization effects. Furthermore, microscopy for clinical dental diagnosis is not possible because PLM is only used in transmission geometry and requires tissue biopsy. Optical coherence tomography is another promising technique for dentistry, but, due to expense, our method is more cost-effective and affordable for common clinics.

In this paper, we demonstrate an optical imaging method for dentistry based on optical polarization state analysis (PSA). The proposed method is noninvasive, nonradioactive, and may provide a quantitative assessment, which is essential for further research on the progression of the lesion. In addition, this technique is cost-efficient and, thus, has high potential for clinical use. In the present study, the two-dimensional images and Mueller matrices of three tooth samples were analyzed ex vivo. Two of them are infected with calculus and dental caries, and the other is diagnosed with cracked tooth syndromes. We decomposed the Mueller matrices to derive retardance, diattenuation, and depolarization information [33,34,35]. In this study, we utilized back-reflection geometry for further application to dentistry. The same scheme was used by other researchers and proved useful for turbid tissue measurement in biomedicine [35,36,37]. We applied Lu-Chipman decomposition on Mueller calculus to show the individual contributions of all the optical properties, which is difficult to measure by using other polarization imaging methods. Our results further revealed diattenuation of the diseased tooth, which has never been reported before. This showed the powerfulness of our proposed method. All the results were compared with their visible images and demonstrated identification ability.

## 2. Materials and Methods

### 2.1. Experimental Setup

The optical polarization states analyzing system can be divided into three sections according to their functions, which are operated in sequence and work independently [33]. Figure 1 shows the flowchart and the experimental setup. The first part of this system is the polarization state generator, which is composed of a laser diode with 850 nm wavelength, a linear polarizer, a quarter wave plate, and a half waveplate. The function of this part was to manipulate the polarization states of the output light and to provide a stable light source for the system. First, the light emitted from the laser diode was transmitted through the linear polarizer and turned into linearly polarized light. By tuning the angle of the quarter and half waveplates, we controlled the output polarization states. In our experiment, the six polarization states including vertical, horizontal, 45 degrees, 135 degrees, right-handed, and left-handed polarized lights were used and entered the second part of the system separately. The light emitted from the first part shined on the sample with an incidence angle of 45° and was then scattered, carrying the information of the optical properties of the tooth. The third part of the system is a polarization state analyzer, which collected the backscattered light from the sample in the normal direction and analyzed the collected light to extract the optical properties of the tooth. The polarization state of the backscattered light was analyzed using a TXP5004 platform with a PAX 5710IR1 Polarimeter (Thorlabs, Inc.). All Stokes parameters and Mueller matrices were computed using commercial software LabVIEW. Two one-dimensional mobile translation stages were employed and adjusted manually to enable two-dimensional scanning: one for moving the near-infrared laser diode (LD) transversely and the other for moving the sample vertically (step size: 200 μm). The total scanning area was 2.00 mm × 2.00 mm.

### 2.2. Samples

Figure 2 shows the three samples used in our experiment. All samples were collected from volunteers via clinical extraction and were preprocessed by immersing them in bleach sodium hypochlorite for half an hour to prevent samples from molding. The Institutional Review Board (IRB) at National Yang-Ming University, Taipei, Taiwan (IRB no.100104) approved this study. The data analysis is described in the following section.

### 2.3. Lu-Chipman Decomposition Method

The Mueller calculus is a powerful method that provides the most general and complete description of incoherent polarized lights. By manipulating and analyzing the Stokes vectors, which represent the polarization states of lights, optical properties, such as depolarization, diattenuation, and retardance, can be determined mathematically. According to classic optical theory, an optical system can be regarded as a linear-algebraic equation, which is determined by the properties of optical elements inside the optical system. An optical system gives responses corresponding to the polarization state of an incident light. If we characterize the incident light by its Stokes vector, set *s*, and the light scattered from a sample as a new Stokes vector, set *s’*, the relationship between the incident and outcoming lights is expressed by the equation below.
(1)$$\vec{s\prime }=M\vec{s}$$
where **M** represents a 4 × 4 Mueller matrix, which is associated with the optical characteristics of a sample including the structural direction and polarization properties. However, direct interpretation from a Mueller matrix, which is generally sophisticated, is difficult. Therefore, a common method to analyze Mueller matrices is to decompose the Mueller matrix into several matrices, each of which represents different meanings of optical properties and can be determined by the decomposition methods applied. In our study, we applied Lu-Chipman decomposition [34], which decomposes experimentally measured Mueller matrices into three basic characteristic matrices: a diattenuation matrix **M_D_**, a retardance matrix **M_R_**, and a depolarization matrix **M_Δ_**.
(2)$$M=M_{\Delta }M_{R}M_{D}$$
Attenuation describes the extent of power loss in an absorptive material. However, if the material exhibits orientation selectivity like dichroic crystal, the attenuation coefficient will change according to different polarization states of incident lights. To fully describe the attenuation phenomenon in a material, the two attenuation coefficients (eigenvalues) and the corresponding preferential orientations (eigenpolarizations) should be found, and they can both be derived from the diattenuation matrix. In this case, diattenuation power *D* represents the extent of polarization of attenuation and is defined by:(3)$$D\equiv \frac{T_{q}-T_{r}}{T_{q}+T_{r}}$$
where *T* are the transmittances for two eigenpolarizations. Similar to diattenuation, retardance is another orientation-dependent phenomenon resulting from birefringence properties and represents the phase difference caused by a different refractive index experienced in the fast and slow axis along the light trajectory in an optical material. From the retardance matrix, eigenvalues and eigenstates can be derived. Noted that the eigenpolarization of retardance is not necessarily identical with that of diattenuation at the presence of depolarization. The retardance *R* is defined as:(4)$$R\equiv \delta_{q}-\delta_{r}\to 0\leq R\leq \pi$$
in which *δ* are the phase changes for the eigenpolarizations, which indicate the polarizations of the leading wave and the lagging wave. The retardance can be further divided into linear retardance *R_L_* and circular retardance *R_C_*. For the pilot study, we only focus on the linear retardance, which represents the extent of the linearly retarded phase term and is defined by the equation below.
(5)$$R_{L}\equiv R\sqrt{a_{1}{}^{2}+a_{2}{}^{2}}$$
in which a are two elements from the Stoke vector of the eigenpolarization (1, *a*_1_, *a*_2_, *a*_3_)^T^, which is the leading phase. Lastly, the depolarization describes the extent of polarization states loss of a light passing through a depolarizing medium. The depolarization power, denoted by Δ, can be determined by the three principal depolarization factors, which describe the depolarization capabilities of the depolarizer along its principal axes, and is defined by:(6)$$\Delta \equiv 1-\frac{\left|a|+|b|+|c\right|}{3}\to 0\leq \Delta \leq 1$$
in which 1-|*a*|, 1-|*b*| and 1-|*c*| represent the depolarization capabilities along its principal axes.

Stepwise, the three characteristic parameters mentioned above can be derived from the measured Mueller matrix. By utilizing the symmetricity of a diattenuation matrix, the diattenuation vector can be directly extracted from the measured Mueller matrix and used to calculate the diattenuation power. Secondly, since a retardance matrix is a unitary matrix, we can separate it with a depolarization matrix simply by multiplying a transposed matrix and, subsequently, obtain both retardance and depolarization power. For more detail on the Lu-Chipman decomposition calculation procedure, please review Reference [34].

Using these basic characteristic matrices obtained via Lu-Chipman decomposition enables us to quantize the three types of optical properties. By scanning samples two-dimensionally, three images corresponding to the three optical characteristics can be produced to facilitate interpreting the results meaningfully.

### 2.4. Data Processing

Figure 3 shows the data flow diagram depicting the decomposition process. We measured 256 Stokes vectors for each sampling point and averaged the data for noise reduction. Subsequently, we adjusted the input polarization states as mentioned in the first paragraph of this section and repeated the same procedure to acquire the corresponding Stokes vectors six times. By using Equation (1), we calculated the Mueller matrix, which represents the optical system of the sample, and applied the Lu-Chipman decomposition to obtain the three characteristic matrices shown in Equation (2). Using these three matrices, we quantified the optical properties and generated two-dimensional polarization images. The image processing of the polarimetric analysis was calculated by using the LabVIEW platform.

## 3. Results

### 3.1. Dental Calculus

Figure 4A,B show the visible image and the scanning area of the calculus sample respectively. Clinicians can hardly perform examination and treatment without magnification. Furthermore, the visible image alone is insufficient for identifying calculus due to the poor color contrast between the infected and the normal parts.

Figure 5A,B show our results of the linear retardance and diattenuation images of the calculus sample, which depicts the orientation-dependent characteristics of the imaging area. These two images are in good alignment with the visible image and less observed in the Mueller matrix entries images, which shows great contrast between the calculus and the normal part. Compared with the healthy enamel surface, the calculus exhibits lower linear retardance and higher diattenuation, which result from the differences of birefringence properties and orientation selectivities between the enamel and the calculus. Unlike linear retardance and diattenuation, instead of representing the orientation-dependent characteristics, depolarization power describes the extent of polarization state loss and has no relation with orientations. The difference of depolarization power between the calculus and healthy enamel was tiny and not significant over the entire scanning area. Therefore, the depolarization image result is not shown in this paper. (We do not show the depolarization images of the other samples for the same reason.)

### 3.2. Dental Caries

The cause of caries is bacterial breakdown of hard tissues of teeth. If the demineralization rate resulting from acids produced by sugar breakdown is greater than the remineralization rate of the oral cavity, caries occur. The surface of the enamel is first invaded and becomes dark. Figure 6A,B is the visible image and the scanning area of the dental caries sample. In contrast to the calculus sample, dental caries can be observed more easily.

Figure 7A,B show the analysis results of the dental caries tooth sample. As we can see from the figure, the caries, which are similar to calculus, features overall lower linear retardance and higher diattenuation than the normal tissue. The caries can be distinguished more clearly in a diattenuation image than in a linear retardance image whereas the calculus can be identified from either. We can even observe opposite results showing up in the linear retardance image in the caries area. To sum up, the diattenuation image exhibits higher consistency with the visible image when compared to the linear retardance image.

### 3.3. Cracked Tooth Syndrome

The visualization of the tooth site with cracked tooth syndrome is vital for a clinical diagnosis. Currently, although an optical method like transillumination has been investigated for localizing a lesion, this treatment approach still relies on clinicians’ judgement by using bare eyes. Figure 8A,B show sample pictures of the cracked tooth syndrome and its scanning area. Such a small fissure is difficult to diagnose and monitor without a microscope or any other dental assistant tools.

Figure 9A,B show the linear retardance image and diattenuation image results of the cracked tooth syndrome sample. These two images both show good ability to differentiate the cracked site from the normal tissue. The infected area exhibits generally lower linear retardance and higher diattenuation, whose tendencies are the same as those of the other two dental hard tissue diseases. However, there are some areas at the left lower corner of the linear retardance image where the tooth is not cracked showing a higher value. The specificity is, therefore, decreased. Nonetheless, the diattenuation image shows the highest accuracy among all decomposition images to distinguish the cracked tooth.

## 4. Discussion

Since the enamel exhibits a unique birefringence property, the appearances of the differences of diattenuation and linear retardance, which represent the extent of orientation-dependent property, between infected and healthy parts meet our expectation. As for the results of cracked tooth syndrome, the signal differences between the cracked part and the normal part are more likely arising from the mechanical structure of a fissure. Since a fissure exhibits anisotropic fracture section, or say interface, the polarization state is partially deattenuated, according to the incidence angle. We speculated that the small linear retardance is likely induced by demineralization after the formation of cracks. Our experimental results indicated that all the dental hard tissue diseases exhibit higher diattenuation and overall lower linear retardance when compared to the normal tissue whereas the changes of depolarization are not significant. It is understandable that the infected area has lower linear retardance than the healthy part because of the damage of the enamel nanostructure, which leads to the loss of its birefringence properties. Although there are some adverse results and an unexpected raising value appears in infected and healthy areas, we can still roughly differentiate the lesion via the tendency of linear retardance images. The unexpected results may arise from local singularities such as the lesion boundary or other structural factors.

In contrast to the linear retardance, the diattenuation images exhibit a high signal-to-noise ratio and have higher capability to identify the infected area. However, the higher diattenuation of the infected area implies that dental calculus and caries, surprisingly, exhibit orientation selectivity as well. To say it more specifically, both dental calculus and caries can be regarded as di-attenuators and have specific preferential orientation, according to our experimental results. Previous study on the nanostructure of carious tooth assessed by small-angle X-ray scattering (SAXS) [38] concluded that the nanostructural orientation of the tooth does not change when the tooth is infected with dental caries, which implied that, when invaded by dental caries, enamel only loses in its compositions but not in its orientations. Our image further indicated that the infected tooth exhibits a diattenuation property, which may result from an unequal rate of the demineralization process due to enamel nanostructure. As for dental calculus, microscopic studies also reported that two kinds of mineralization centers, designated with A and B [39], can be identified by polarized light. From our image result, we speculated that diattenuation of dental calculus may arise from the mineralization centers mentioned above. Nonetheless, up to now, there are no relevant studies linked with the pathological observation of our new findings directly. To our knowledge, it was the first time that dental caries and calculus were inspected to exhibit diattenuation properties, and, thus, more fundamental experiments should be further carried out to verify and to link with pathology.

In the present study, the incidence angle and received angle play important roles in the final polarization state analysis because we employed back-reflection geometry in our experimental setup. If it is not both a normal incidence and normally received, additional phases would be introduced in the depolarized optical system. Therefore, linear terms and circular terms would tangle mutually. In our system setup, we held the incidence angle and the received angle at 45° and 0°, respectively, to detect scattered light. Despite the fact that the influence of the additional phase was not taken into account in our polarization state analysis procedure, the indices of retardance, diattenuation, and depolarization evaluated the overall consequences, which included linear terms and circular terms, and were, thus, free from the influence of the additional phase. In our results, only linear retardance would be influenced. Although the variance of the measured polarization state is also influenced by defocusing in reality, it is still acceptable within Rayleigh length, which is up to 3 mm in our system. Therefore, the impact of defocusing can be negligible. Another issue is that sodium hypochlorite, which is previously reported responsible for the deproteinization of the enamel surface [40,41], was used in our sterilization procedure. The enamel surface will be sterilized and altered in compositions as well as morphology at the same time, which leads to ambiguity of our results. Nonetheless, in the present study, we only focused on the applicability of the proposed method in detecting dental hard tissue diseases, and the immersive sterilization is expected to cause universally equal changes in abnormal or normal areas and has no influence on the qualitative conclusions of our experiments.

Although we only compare abnormal areas with relatively healthy areas in the same sample in this study since intact tooth samples are not available from clinical extractions, we think it is enough and proper to distinguish abnormal parts from normal parts in our experimental manner because differences between the normal and the abnormal parts can be compared only within one sample in consideration of individual differences.

## 5. Conclusions

In this paper, we demonstrated the feasibility of the optical polarimetric detection method for assessing dental hard tissue diseases including dental calculus, caries, and cracked tooth syndrome. We constructed a back-reflection polarization states analysis system and conducted the experiments ex vivo. By using the Lu-Chipman decomposition, the experimentally acquired data were transformed into three meaningful and quantitative values of optical properties. The results were presented in 2D images by using mobile translation stages, which exhibit a high ability to distinguish lesions in comparison with visible images. Our results show that whatever kinds of dental hard tissue diseases are infected with, the lesion parts all exhibit high diattenuation and lower linear retardance than the normal areas whereas the depolarization component varied little. From these imaging results, we concluded that the dental caries and dental calculus can be regarded as di-attenuators due to their nanostructural crystalline orientations whereas the results of cracked tooth syndrome might arise from superficial mechanical structures. The cause of diattenuation in dental calculus and dental caries should be further investigated in more fundamental experiments. Despite the limited sample size, according to our image results, it is clearly shown that the proposed method has the power to differentiate abnormal areas.

Our method provides not only quantitative indices but also diagnosing ability. The proposed optical polarization imaging system for oral medicine is noninvasive and nonradioactive. Moreover, polarization imaging is an inexpensive approach for detecting dental hard tissue diseases and demonstrated the potential for application in oral medicine real-time diagnosis. We anticipate that the polarization method can also be an assistant or even guiding tool when clinicians are performing clinical surgeries. Currently, only ex vivo samples are studied, and the image resolution is restricted by the step size of the manually adjusted translational stages. Notwithstanding, these issues can be improved in the following manner. In future studies, we will first collect more samples to validate our preliminary results and carry out the experiment in vivo. Additionally, we will manage to substitute two stages with a charge-coupled device to shorten the measuring time and to increase the image resolution. Furthermore, the detector can be miniaturized for oral endoscopy or for developing new tools for detecting subgingival dental calculus. We are looking forward to the proposed analysis method expediting clinical examinations as well as the development of research on the nanostructural progression of oral diseases.

## Figures and Tables

**Figure 1 sensors-19-04971-f001:**
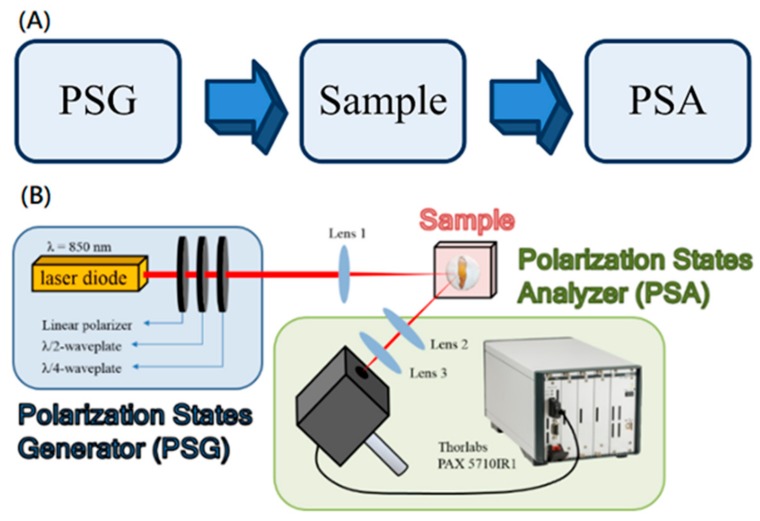
The system diagram. (**A**) The whole system can be divided into three sections according to their functions. These three sections of the system are operated in sequence and work independently. (**B**) The figure shows the experimental setup of back-reflection polarization states analyzing system. The polarization states generator (PSG) is composed of a laser diode with 850 nm wavelength, one linear polarizer, and two waveplates. The light escaping from the PSG is then focused to obtain single-spot data on the sample, which is collected by the detector. LP: linear polarizer. HWP: half waveplate. QWP: quarter waveplate.

**Figure 2 sensors-19-04971-f002:**
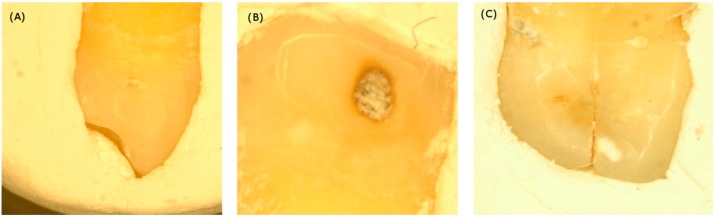
The visible images of (**A**) calculus, (**B**) dental caries, and (**C**) cracked tooth syndromes samples. All samples were collected from volunteers via clinical extraction and were preprocessed by immersing them in bleach for half an hour to prevent samples from molding.

**Figure 3 sensors-19-04971-f003:**
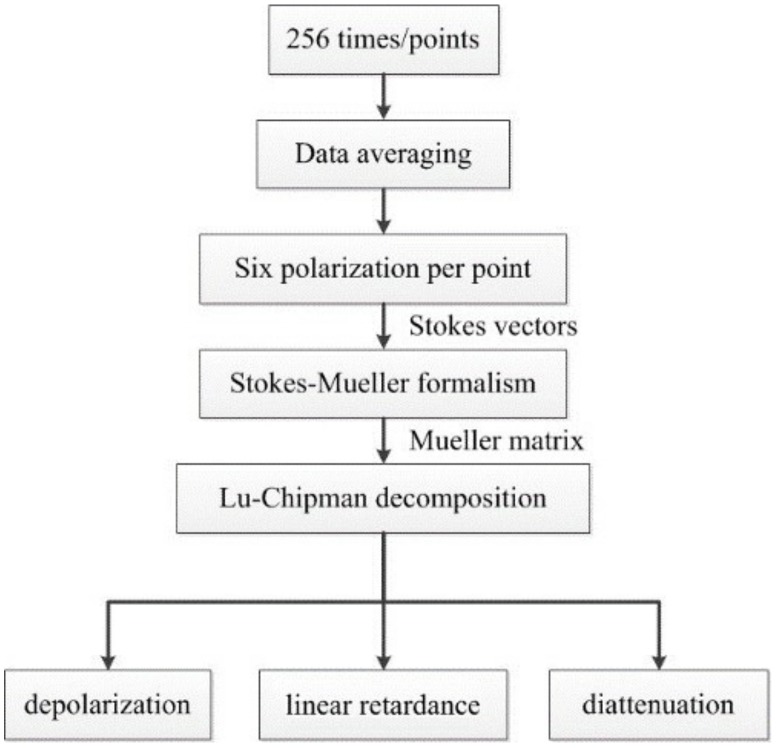
We measured 256 Stokes vectors for each sampling point and averaged the data for noise reduction. By adjusting the input polarization state six times, the corresponding Stokes vectors were acquired. By using Equation (1), we calculated the Mueller matrix and applied the Lu-Chipman decomposition to obtain three two-dimensional imaging results for each sample.

**Figure 4 sensors-19-04971-f004:**
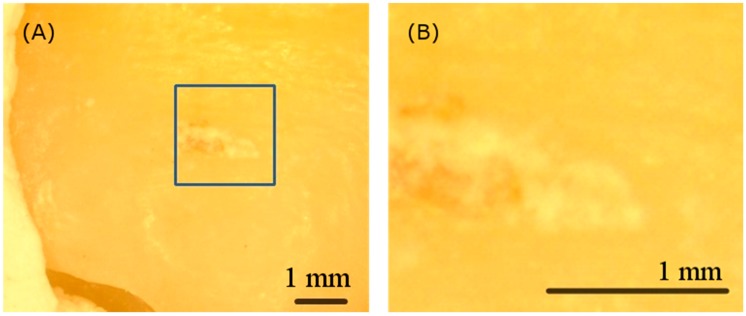
(**A**) The visible image of the calculus tooth sample and (**B**) the zoom-in scanning area image captured by the microscope. The visible image is insufficient for clinicians to identify calculus without magnification.

**Figure 5 sensors-19-04971-f005:**
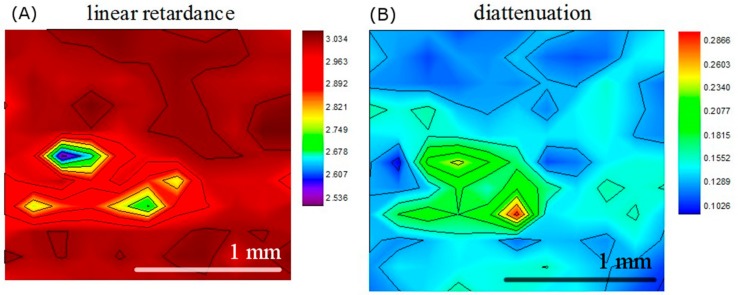
(**A**) Linear retardance image and (**B**) diattenuation image of the calculus tooth sample. Both image results show good ability to identify the calculus.

**Figure 6 sensors-19-04971-f006:**
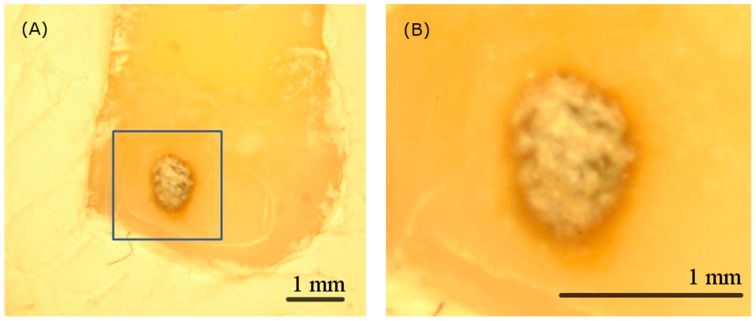
(**A**) The visible image of the dental caries tooth sample and (**B**) the zoom-in scanning area image captured by the microscope.

**Figure 7 sensors-19-04971-f007:**
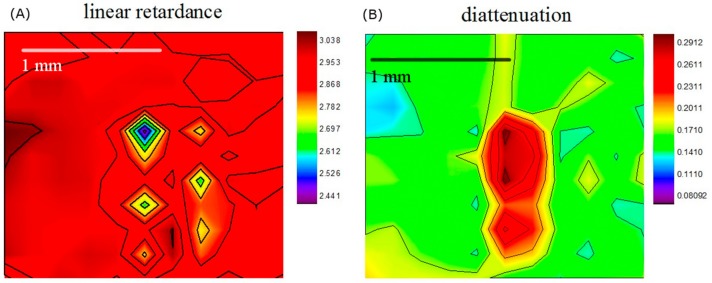
(**A**) Linear retardance image and (**B**) diattenuation image of the dental caries tooth sample. We can clearly identify dental caries from both images. However, the diattenuation image has higher accuracy than the linear retardance image.

**Figure 8 sensors-19-04971-f008:**
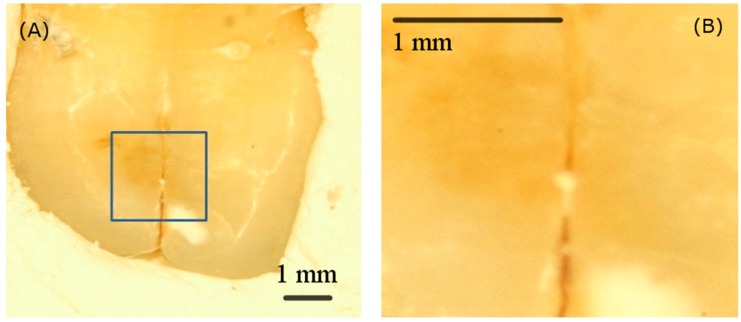
(**A**) The visible image of the cracked tooth syndrome tooth sample and (**B**) the zoom-in scanning area image captured by the microscope. It is difficult for clinicians to diagnose or monitor the site of cracks without a microscope or any other dental assistant tools.

**Figure 9 sensors-19-04971-f009:**
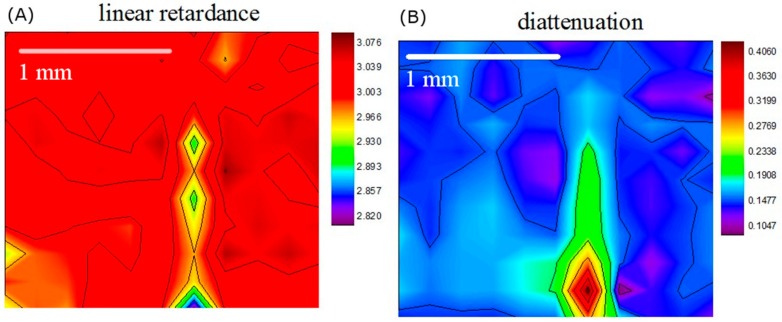
(**A**) Linear retardance image and (**B**) diattenuation image of the cracked tooth syndrome tooth sample. Similar to dental caries, both the linear retardance image and the diattenuation image can identify the site of cracks, but the diattenuation image performs better.

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
