# Peer review of "Optical Polarimetric Detection for Dental Hard Tissue Diseases Characterization"

_sensors, 2019, doi:10.3390/s19224971_

Round 1

Reviewer 1 Report

In the current manuscript, the authors report a demonstrate an optical polarization imaging method for predicting the diagnostic potential of dental calculus, caries, and cracked tooth on explanted specimens obtained form dental extractions. The manuscript is well presented and relevant contribution to the journal.

Although the study design and experiments are well-conducted, the study suffers from some limitations that would need to be addressed to strengthen the manuscript.

The present manuscript would be of high interest if the authors demonstrate the results reported in this study being validated in at least a few clinically intact samples. This demonstrates the claimed of the diagnostic application in the manuscript. The manuscript in the current form does not meet the standard. A less ideal method would be to examine the intact tooth without much post-chemical processing in a fabricated tooth model. 

The sample size is too low. I suggest the authors examine additional specimens in blinded study design to predict the accurate diagnosis and correlate with the clinical findings or any variables. 

The authors discuss the application in soft tissues but no evidence is shown. I suggest to rephrase this application with caution or remove.

The authors may consider describing any limitations of the study in the discussion.

Reviewer 2 Report

This paper presents a unique method of detecting dental hard tissue disease which is of great interest to the scientific community. I feel the manuscript could be improved in the following ways

The first paragraph of the introduction talks about terms such as Tomes' process, ameloblasts etc. for which no reference is provided other than [1]. This opening paragraph needs more references.  The resolution of linear retardance and diattenuation seem to be not so good, and probably that's the reason why there are so many sharp corners. It would be better if the resolution could be improved  The authors might consider providing a comparison of their proposed method with the existing ones in a tabular format in the Discussion section. That would greatly enhance the quality of this manuscript
